# The PAK1-Stat3 Signaling Pathway Activates *IL-6* Gene Transcription and Human Breast Cancer Stem Cell Formation

**DOI:** 10.3390/cancers11101527

**Published:** 2019-10-10

**Authors:** Ji-Hyang Kim, Hack Sun Choi, Su-Lim Kim, Dong-Sun Lee

**Affiliations:** 1Interdisciplinary Graduate Program in Advanced Convergence Technology & Science, Jeju National University, Jeju 63243, Korea; seogwi12@naver.com (J.-H.K.); ksl1101@naver.com (S.-L.K.); 2Faculty of Biotechnology, College of Applied Life Sciences, Jeju National University, SARI, Jeju 63243, Korea; choix074@jejunu.ac.kr; 3Subtropical/Tropical Organism Gene Bank, Jeju National University, Jeju 63243, Korea; 4School of Biomaterials Science and Technology, College of Applied Life Science, Jeju National University, Jeju 63243, Korea; 5Practical Translational Research Center, Jeju National University, Jeju 63243, Korea

**Keywords:** PAK1, cancer stem cell, Stat3, IL-6, mammosphere, JAK2

## Abstract

Cancer stem cells (CSCs) have unique properties, including self-renewal, differentiation, and chemoresistance. In this study, we found that p21-activated kinase (PAK1) inhibitor (Group I, PAK inhibitor, IPA-3) and inactivator (ivermectin) treatments inhibit cell proliferation and that tumor growth of PAK1-knockout cells in a mouse model is significantly reduced. IPA-3 and ivermectin inhibit CSC formation. PAK1 physically interacts with Janus Kinase 2 (JAK2), and JAK2 inhibitor (TG101209) treatment inhibits mammosphere formation and reduces the nuclear PAK1 protein level. PAK1 interacts with signal transducer and activator of transcription 3 (Stat3), and PAK1 and Stat3 colocalize in the nucleus. We show through electrophoretic mobility shift assay (EMSA), chromatin immunoprecipitation (ChIP), and reporter assays that the PAK1/Stat3 complex binds to the *IL-6* promoter and regulates the transcription of the *IL-6* gene. Inhibition of PAK1 and JAK2 in mammospheres reduces the nuclear pStat3 and extracellular IL-6 levels. PAK1 inactivation inhibits CSC formation by decreasing pStat3 and extracellular IL-6 levels. Our results reveal that JAK2/PAK1 dysregulation inhibits the Stat3 signaling pathway and CSC formation, the PAK1/Stat3 complex regulates *IL-6* gene expression, PAK1/Stat3 signaling regulates CSC formation, and PAK1 may be an important target for treating breast cancer.

## 1. Introduction

Breast tumors show intratumor heterogeneity that is responsible for chemoresistance and tumor progression [1]. Tumors contain heterogeneous cell populations. Cancer stem cells (CSCs), stem cell-like cells, comprise a small population in tumors and play an important role in cancer relapse, metastasis and chemoresistance [2,3,4]. CSCs are more resistant than cancer cells to conventional therapy, radiotherapy, chemotherapy and hormone therapy [5,6,7]. The most common biomarkers of breast CSCs are CD44 and ALDH1 [8]. Notch and hedgehog signaling play a role in breast CSC survival, and these pathways are targets for breast CSC elimination without ablating normal cell function [9]. The regulation of miRNAs is associated with carcinogenesis and chemoresistance in breast CSCs. The overexpression of let-7 miRNA is involved in breast CSC formation, and miRNA-based therapy is still needed to regulate breast cancer stemness [10]. Breast CSC research enables an enhanced understanding of the nature of breast cancer and CSCs. The most important proteins in all cancers are NF-κB and Stat3. The transcription factors NF-κB and Stat3 are activated in many cancers and play an essential role in CSC formation [11,12].

The p21-activated kinases (PAKs) are serine/threonine kinases. PAKs consist of PAKs 1-6 and play roles in cell growth, cytoskeletal organization, and cell migration. PAK1 and PAK4 proteins are associated with cancer tumorigenesis [13]. PAK1 and PAK4 levels are highly elevated in many cancers, and the other members of the PAK family are also linked to cancer [14]. PAKs in adult cancer tissue may be used as a diagnostic biomarker [13]. The PAK1 pathway affects nuclear events, such as transcriptional activation [15,16] and regulates the transcription of target genes [17,18,19,20]. PAK4 maintains cancer stem cell phenotypes through Stat3 signaling [21]. PAK3 promotes CSC phenotypes through Akt-GSK3β-β-catenin signaling in pancreatic cancers [22]. 

In this study, we examined the function of PAK1 in breast CSC formation. We found that PAK1 interacts with JAK2 and Stat3, and then activates Stat3 signaling to support mammosphere formation and stemness of breast cancer cells. PAK1 induces *IL-6* transcription and in vivo tumor growth. These data show that PAK1 may be a cancer target and CSCs may be killed by targeting PAK1. 

## 2. Results

### 2.1. PAK1 Is Highly Expressed and Evenly Distributed in the Cytosol and Nucleus in Breast CSCs

To examine the function of PAK1 in breast CSC formation, we cultured and isolated mammospheres derived from breast cancers. As PAK1 is strongly expressed in squamous non-small cell lung cancer (NSCLC) cells [23], we assessed the protein level of PAK1 in breast cancer cells and CSCs by immunoblot analysis. We found that the level of PAK1 protein is markedly higher in mammospheres than in cancer cells (Figure 1A). The densitometry analysis of PAK1 bands showed that the expression of PAK1 was increased two- and 10-fold in breast CSCs (Figure 1B). These data show that PAK1 is highly expressed in breast CSCs and is evenly distributed in the cytosol and nucleus in breast CSCs (Figure 1C).

### 2.2. PAK1 Inhibition Effectively Inhibits Proliferation, Migration, and Colony Formation in Cell Lines and Tumor Growth in an In Vivo Model Using PAK1-Knockout HAP1 Cells

We investigated the antiproliferation effect of the PAK1 inhibitors IPA-3 and ivermectin [24] on human breast cancers. PAK1 inhibitor treatment reduced the proliferation of breast cancer cells (Figure 2A,B). Apoptosis in breast cancer cells was induced by ivermectin at a concentration of 20 μM (Figure 2C). Ivermectin induced caspase 3/7 activity in breast cancer cells (Figure 2D). IPA-3 and ivermectin treatment reduced cell migration and colony formation (Figure 2E,F). Our data indicate that PAK1 is essential for the regulation of cell proliferation, migration, and colony formation. As PAK1 regulates cancer cell proliferation, we examined whether PAK1 regulates tumor growth using a xenograft tumor model. The expression level of PAK1 protein in PAK1-knockout and wild-type HAP1 (haploid human) cells was assessed by immunoblot analysis using anti-PAK1. PAK1-knockout HAP1 cells had no PAK1 protein signal (Figure 2G). The tumor volume in the PAK1-knockout HAP1 cell-injected mice was smaller than that in the HAP1 cell-injected mice (Figure 2H,I). Additionally, the tumor weights in the PAK1-knockout HAP1 cell-injected group were lower than those in the HAP1 control cell-injected group (Figure 2I). Tumors derived from nude mice injected with HAP1 cells had higher PAK1 protein levels than tumors derived from nude mice injected with PAK1-knockout HAP1 cells (Figure 2J). These results show that PAK effectively regulates tumorigenicity in a xenograft model. 

### 2.3. Inhibition of PAK1 Suppresses Mammosphere Formation

CSC self-renewal maintains the undifferentiated state [25]. We examined the role of PAK1 in mammosphere formation. To evaluate whether PAK1 inhibitor, IPA-3 and PAK1 inactivator, ivermectin can inhibit mammosphere formation, we treated mammospheres with IPA-3 and ivermectin. As shown in Figure 3A,B, IPA-3 and ivermectin treatment inhibited the formation of mammospheres. Mammosphere numbers declined by 55% to 94%, and the size of the mammospheres decreased (Figure 3A, B). To examine the effects of ivermectin, we assessed the level of PAK1 protein. Ivermectin treatment reduced the protein level of PAK1. Next, we examined whether PAK1 plays a significant role in breast CSC stemness using siRNA-induced silencing of PAK1 expression. PAK1 silencing in breast cancer cells decreased mammosphere formation (Figure 3C,D).

### 2.4. Ivermectin Treatment Reduces the CD44^+^/CD24^−^ and ALDH-expressing Breast Cancer Cell Population

We treated cancer cells with ivermectin and analyzed the CD44^+^/CD24^−^-expressing subpopulation to examine the effect of ivermectin. Ivermectin decreased the CD44^+^/CD24^−^-expressing cancer cell population from 14.2% to 1.6% (Figure 3E). To examine the effect of ivermectin on ALDH (aldehyde dehydrogenase)-positive cells, we performed an ALDEFLUOR assay. Ivermectin treatment reduced the ALDH-positive cell population from 1.6% to 0.8% (Figure 3F). Therefore, ivermectin treatment reduces breast CSC hallmarks (CD44^+^/CD24^−^-expressing cells and ALDH-expressing cells).

### 2.5. PAK1 Physically Interacts with JAK2, and the PAK1-JAK2 Interaction Regulates CSC Formation in Human Breast Cancer

The JAK2/Stat3 signaling pathway is essential for human breast CSC growth [1]. As PAK1 is important for CSC formation, we examined the relationship between JAK2 and PAK1. To test this hypothesis, protein extracts were immunoprecipitated with specific antibodies, and JAK2 and PAK1 were identified using western blotting. PAK1 but not control IgG coimmunoprecipitated with JAK2 (Figure 4A). This result suggests that PAK1 and JAK2 interact in vivo. We examined the function of JAK2 in CSC formation using siRNA-mediated silencing of JAK2 expression. JAK2 silencing inhibited the formation of breast cancer mammospheres (Figure 4B). TG101209 is a selective JAK2 kinase inhibitor [26]. To examine whether TG101209 inhibits the formation of tumorspheres, we treated mammospheres derived from breast cancer cells with TG101209. Figure 4D shows that TG101209 treatment inhibited mammosphere formation. Mammosphere numbers declined by 40%, and the size of the mammospheres was reduced (Figure 4D).

### 2.6. JAK2 Inhibition Reduces PAK1 Nuclear Localization

To understand the inhibition of mammosphere formation by TG101209, we checked the levels of cytosolic and nuclear PAK1 in mammospheres treated with TG101209. We observed that the total protein level of PAK1 was unchanged and that nuclear PAK1 protein was decreased by TG101209 treatment (Figure 4E,F). Furthermore, immunofluorescence analysis of MDA-MB-231 cells indicated that TG101209-treated cells expressed lower levels of nuclear PAK1 than untreated cancer cells (Figure 4G). Our observations suggest that the reduction in nuclear PAK1 is important for mammosphere formation in breast cancer lines.

### 2.7. PAK1 Physically Interacts with Stat3 and Regulates Stat3 Phosphorylation in Human Breast Cancer Lines

Because PAK1 interacts with JAK2, we sought to identify PAK1-interacting proteins in breast cancer cells to determine the molecular function of PAK1. Because Stat3 is an essential factor in CSC formation [12], we checked the interaction between PAK1 and Stat3. The results in Figure 5A show that PAK1 coimmunoprecipitated with Stat3 in breast cancer cell lines, suggesting that two proteins interact in vivo (Figure 5A). The immunofluorescence analysis with anti-PAK1 and anti-Stat3 in cancer cells indicated that PAK1 and Stat3 colocalize in the nucleus (Figure 5B). Our observations suggest that PAK1 and Stat3 interact and colocalize in the nucleus in breast cancer cell lines. To investigate the cellular function of downregulated PAK1 expression, we examined Stat3 phosphorylation in mammospheres treated with ivermectin. Compared to the control treatment, ivermectin treatment reduced the pStat3 level (Figure 5C). The immunofluorescence analysis with anti-PAK1 and anti-Stat3 in breast cancer cells with/without ivermectin treatment showed that colocalization of PAK1 and Stat3 in the nucleus was reduced by ivermectin treatment (Figure 5D).

### 2.8. JAK2 Regulates the PAK1/Stat3 Signaling Axis

JAK2 interacts with PAK1 and regulates the nuclear PAK1 level and mammosphere formation. PAK1 interacts with Stat3 and regulates the nuclear Stat3 level and mammosphere formation. Therefore, we examined the role of the JAK2/PAK1/Stat3 signaling axis in mammosphere formation. We examined the JAK2/PAK1/Stat3 signaling axis by using the JAK2 inhibitor TG101209 and immunofluorescence analysis using anti-PAK and anti-Stat3. Compared to vehicle treatment, TG101209 treatment markedly reduced the level of nuclear PAK1 and Stat3 proteins. Additionally, colocalization of nuclear PAK1 and Stat3 was reduced in TG101209-treated cells compared to vehicle-treated cells (Figure 6A). We examined cellular and nuclear pStat3 levels in JAK2 inhibitor TG101209-treated cells. We found that Stat3 phosphorylation was reduced after TG101209 treatment (Figure 6B). These results show that JAK2 regulates the PAK1/Stat3 signaling pathway.

### 2.9. Involvement of PAK1 Cotranscription Factor in Stat3-Mediated *IL-6* Expression Upregulation

Though PAK1 lacks transcriptional activity, our data show that PAK1 and Stat3 induce *IL-6* gene expression (Figure 7D). We hypothesized that PAK1 could be recruited to the *IL-6* gene promoter with Stat3, which could induce direct stimulation of *IL-6* transcription. We investigated the direct binding of PAK1 and Stat3 to a Stat3 binding probe by electrophoretic mobility shift assay (EMSA) (Figure 7A,B). We tested Stat3-specific DNA binding in nuclear extracts from mammospheres using an IRDye 800-labeled Stat probe that binds Stat3. We found that Stat3 in the nuclear extracts bound to the IRDye 800-labeled Stat probe (indicated by arrow) (Figure 7A,B). The specificity of the Stat3/IRDye 800-labeled Stat3 probe was confirmed by using a 10-fold concentration of a self-competitor oligo (Figure 7A,B, lane 3) and a 10-fold concentration of a mutated Stat oligo (Figure 7A,B, lane 4). In Figure 7B, the major bands indicated by arrows are the Stat3/DNA complex. A major supershifted band was observed using anti-pStat3 (Figure 7B, lane 5), anti-PAK1 (Figure 7B, lane 6) and a combination of anti-pStat3 and anti-PAK1 (Figure 7B, lane 7). These results show that Stat3 is a major transcription factor and that PAK1 is recruited by Stat3. To assess whether Stat3 and PAK1 regulate endogenous *IL-6* gene regulation, a ChIP (chromatin immunoprecipitation) assay was carried out to investigate the presence of the Stat3/PAK1 complex on the human *IL-6* promoter. As shown in Figure 7C, the precipitated DNA fragments containing the 191-bp region covering the Stat3 binding site of the human *IL-6* promoter were amplified with specific primers. ChIP-PCR products were detected in breast cancer cells by using a Stat3 or PAK1 antibody, but not detected with the control IgG (Figure 7C). To assess whether the Stat3/PAK1 complex interacts with the promoter region of the *IL-6* gene, Re-ChIP was performed using the indicated antibodies. We found that PAK1 and Stat3 proteins are recruited to the *IL-6* promoter region (Figure 7C), demonstrating the interaction of both proteins at the *IL-6* promoter. To evaluate whether transient expression PAK1 and Stat3 results in an increase in endogenous *IL-6* gene expression, RT-qPCR of the *IL-6* gene was performed by using total RNAs derived from breast cancer cells transfected with pPAK1 (1 μg), pStat3 (1 µg), and pPAK1 plus pStat3 using the pcDNA3.1 vector as an internal control. As shown in Figure 7D, Stat3 and PAK1 overexpression induced *IL-6* transcription. These results indicate that Stat3 and PAK1 play important roles in *IL-6* gene expression. 

To evaluate whether PAK1 is crucial for breast CSC formation, the effect of ivermectin on Stat3/IL-6 signaling was assessed. Ivermectin treatment decreased the phosphorylation levels of Stat3 and PAK1 (Figure 7E). However, ivermectin treatment did not decrease nuclear p65 levels (Figure 7E). Additionally, we assayed Stat3 binding in ivermectin-treated nuclear protein extracts using the IRDye 800-labeled Stat probe. Ivermectin treatment inhibited Stat3 binding (Figure 7F, lane 3). The binding specificity of the IRDye 800-Stat probe was confirmed by using a 10-fold concentration of the competitor oligo (Figure 7F, lane 4) and 10-fold concentration of the mutated Stat oligo (Figure 7F, lane 5). Extracellular IL-6 is known to play an essential role in CSC formation [27]. To assess the extracellular IL-6 level, we checked the IL-6 level in the culture media using an IL-6 antibody. RT-qPCR and western blot data showed that ivermectin treatment reduced IL-6 transcripts (Figure 7G) and extracellular IL-6 protein levels (Figure 7H). The control was the equivalent number of mammospheres with/without ivermectin treatment (Figure 7H). After ivermectin treatment, the cytokine profiling data showed that ivermectin reduced the level of extracellular IL-6 (Figure 7I).

These results show that PAK1/Stat3/IL-6 signaling is very important for regulating mammosphere formation in breast cancer.

### 2.10. PAK1 Regulates CSC-Specific Gene Expression and Mammosphere Proliferation

RT-qPCR was used to examine whether ivermectin treatment inhibits CSC-specific gene expression. Ivermectin reduced Nanog, CD44, Sox2, Oct4, and c-Myc gene expression levels in mammospheres (Figure 8A). To examine whether ivermectin treatment blocks mammosphere proliferation, mammospheres were cultured in media with ivermectin, and the mammosphere cell numbers were counted. Ivermectin treatment inhibited mammosphere growth. This result showed that ivermectin treatment led to a dramatic arrest of mammosphere growth (Figure 8B). These results suggest that treatment with ivermectin, which inactivates PAK1, inhibits the stemness and proliferation of CSCs. 

## 3. Discussion

The key features of breast tumors are their heterogeneity and complex nature [28]. CSCs, similar to normal stem cells, are involved in the chemoresistance of cancers [29]. Breast CSCs have been shown to be resistant to cancer therapy [6]. 

Accumulating data show that PAK1, PAK3, and PAK4 regulate cell proliferation, motility, and angiogenesis in several types of cancer cells [30]. For the first time, we show that PAK1 plays an important role in breast CSC growth via JAK2/PAK1/Stat3/IL-6 signaling. The main reason for chemoresistance is the CSC subpopulation in the bulk tumor. We found that PAK1 silencing inhibited mammosphere formation, reduced the CD44^+^/CD24^−^ CSC population and reduced ALDH activity. Therefore, these results indicate that it is universally associated with the CSC phenotype. Human breast cancer cells express both CD24 and CD44 [31], but MDA-MB-231 cancer cells expressed a high level of CD44 and a very low level of CD24. Our in vitro system, MDA-MB-231 cells and mammospheres derived from MDA-MB-231 cells, have a limitation that the cell model being used may not reflect the actual status of breast cancer cells.

Using several experiments involving immunoprecipitation, immunofluorescence, and JAK2 inhibition, we found that JAK2 interacts with PAK1 and inhibits CSC formation. The JAK2/Stat3 signaling pathway is required for human breast CSCs [1]. Furthermore, PAK1 silencing not only reduced mammosphere formation but also reduced CSC-specific markers. 

Using various experimental methods, we found that PAK1 interacts with Stat3 and that PAK1 and Stat3 colocalize in the nucleus. PAK1 inhibition induced Stat3 dephosphorylation (Figure 5). CSC formation is regulated by extracellular IL-6 and IL-8 [32]. The JAK/Stat3 pathway in human breast cancer drives tumorigenesis and metastasis. To investigate the direct recruitment of Stat3 to the promoter region of the *IL-6* gene in breast cancer cells, we performed EMSA and ChIP assays. We found that Stat3 binds to the promoter of the *IL-6* gene and induced *IL-6* gene expression. Furthermore, PAK1 also binds to the *IL-6* promoter (Figure 7). We showed that PAK1 and Stat3 can be co-recruited to the same region of the *IL-6* promoter using double-ChIP with anti-PAK1 and anti-Stat3. Similar approaches have shown that c-JUN and PAK1 regulate tissue factor transcription and that the two proteins are co-recruited to the promoter region of the tissue factor gene [20]. PAK1 confers β-catenin-mediated stemness in non-small cell lung cancer and induced cancer stem cell marker of colorectal cancer [33,34]. PAK-1 mediated stem-like phenotype through NF-kB/IL6 activation in renal cell carcinoma [35]. Therefore, PAK1 play an important role in cancer stemness. We discussed the correlation of PAK1/IL-6 and Stat3/IL-6 in breast cancer patient from the publicly available dataset such as TCGA (The cancer genome atlas) (Appendix A).

In conclusion, our study shows that Stat3 and PAK1 regulate *IL-6* gene expression and that the Stat3/PAK1 complex is recruited to the *IL-6* promoter and induces transcription of the *IL-6* gene. The Stat3/PAK1 complex is essential for *IL-6* gene expression and breast cancer stemness. PAK1 targeting is important for the management of bulk tumor cells and CSCs.

## 4. Materials and Methods

### 4.1. Cell Lines and Media

MCF-7 and MDA-MB-231 cells were grown in Dulbecco’s modified Eagle’s medium (Gibco, Thermo Fisher Scientific, Waltham, MA, USA) supplemented with 10% (V/V) FBS (Thermo Fisher Scientific), and 1% penicillin/streptomycin in a 5% humidified CO_2_ incubator at 37 ℃. The MCF-7 and MDA-MB-231 breast cancer cells were grown at 3.5 × 10^4^ or 1 × 10^4^ cells per well in ultralow adherent plates with MammoCult^TM^ medium (STEMCELL Technologies, Vancouver, BC, Canada) supplemented with heparin and hydrocortisone and were cultured in a 5% humidified CO_2_ incubator at 37 °C. At seven days of culture, a six-well plate was scanned, and counting of mammospheres was achieved using the NICE program [36]. The mammosphere formation was determined by the mammosphere formation efficiency (MFE; %) as previously described [37].

### 4.2. Antibodies, siRNA, and Plasmids

Anti-PAK1, anti-pStat3 (Y705), and anti-JAK2 antibodies were obtained from Cell Signaling (Danvers, MA, USA). Anti-p65, anti-Stat3, anti-β-actin, and anti-Lamin b were obtained from Santa Cruz Biotechnology (Dallas, TX, USA). Anti-CD44-FITC and anti-CD24-PE antibodies were purchased from BD Pharmingen (Franklin Lakes, NJ, USA). Antibodies for ChIP and EMSA experiments were anti-PAK1 and anti-pStat3. Human PAK1 siRNA and scrambled siRNA were purchased from Bioneer (Daejeon, Korea). The pPAK1 and pStat3 plasmids were purchased from Addgene (Watertown, MA, USA).

### 4.3. Cell Proliferation

We followed a previously reported method [38]. The cancer cells were cultured in 96-well plates with IPA-3 and ivermectin. CellTiter 96^®TM^ Aqueous One Solution (Promega, Madison, WI, USA) was used in accordance with the manufacturer’s protocol, and the OD_490_ was assayed using a plate reader (SpectraMax, Molecular Devices, San Jose, CA, USA), as previously described [39]. 

### 4.4. Caspase-3/7 Assay

Cancer cells were cultured with different concentrations of ivermectin. Caspase-3/7 activity was determined according to the manufacturer’s instructions using the Caspase-Glo 3/7 kit (Promega). Then, 100 μL of Caspase-Glo 3/7 reagent was added to 96-well plates where the cancer cells were cultured and incubated, and the activity was measured by using a GloMax^®^ Explorer plate-reading luminometer (Promega).

### 4.5. Annexin V/PI Assay and Analysis of Cell Apoptosis

Cancer cells were incubated in six-well plates with ivermectin (20 μM) or dimethyl sulfoxide (DMSO). Apoptotic cells were detected by PI and FITC-Annexin V staining according to the manufacturer’s instructions (Becton Dickinson (BD), San Jose, CA, USA). The samples were analyzed by flow cytometry (Accuri C6, BD). 

### 4.6. Clonogenic and Migration Analysis

MDA-MB-231 cells were cultured at 1 × 10^3^ cells in a six-well plate and cultured with different concentrations of IPA-3 and ivermectin in dimethyl sulfoxide (DMEM). The cancer cells were cultured for one week, and cancer cell colonies were counted. Cancer cells were seeded in six-well plates and scratched using a microtip. After washing with DMEM, breast cancer cells were incubated with IPA-3 and ivermectin. We followed a previously described method [39]. 

### 4.7. Flow Cytometry Analysis of CD24 and CD44 Expression

We followed a previously described method [39]. A total of 1 × 10^6^ cells were incubated with FITC-conjugated CD44 and PE-conjugated CD24 (BD) and incubated at 4 °C for 20 min. The breast cancer cells were washed twice with 1XPBS and then analyzed by using a flow cytometer (Accuri C6, BD).

### 4.8. ALDEFLUOR Assay

The ALDH assay was performed using an ALDEFUOR kit (STEMCELL Technologies, Vancouver, BC, Canada). We followed a previously described method [39]. Breast cancer cells were incubated in an ALDH assay buffer at 37 °C for 20 min. The proportion of ALDH-positive cells was analyzed by using a personal flow cytometer (Accuri C6, BD). 

### 4.9. Isolation of RNA and RT-qPCR

Total RNA was isolated, and RT-qPCR was performed using a one-step qRT-PCR kit (Takara, Tokyo, Japan). We followed a previously described method [38]. Primer sequences are shown in Appendix A. The β-actin gene was used as an internal control for RT-qPCR.

### 4.10. Transient Transfection and *IL-6* Gene Expression

Transient transfection was performed as previously described [40]. A total of 1 × 10^6^ cells were cultured in six-well plates before transfection. Two micrograms of plasmid (pPAK1 and pStat3; PAK1 and Stat3 overexpression vector) was transfected into breast cancer cells with Lipofectamine 3000 (Thermo Fisher Scientific, Waltham, MA, USA). After transfection, total RNA was purified using a Mini-BEST kit (Takara). The *IL-6* transcript level was determined using an RT-qPCR kit (Takara). The *IL-6* primers are shown in Appendix A.

### 4.11. Immunofluorescence

Breast cancer cells were fixed with 4% paraformaldehyde for 30 min, permeabilized with 0.5% Triton X-100 for 10 min, blocked with 3% bovine serum albumin (BSA) for 30 min and stained with mouse anti-Stat3, and rabbit anti-PAK1 and followed by secondary anti-mouse Alexa 488 and anti-rabbit Alexa 594 antibodies. We used control-using reduced non-specific signal condition to ensure the specificity of the primary antibodies for immunofluorescence. Finally, slides were mounted in Vectashield with DAPI and visualized with a fluorescence microscope (Lionheart, Biotek, VT, USA).

### 4.12. Immunoblot Analysis

Proteins isolated from breast cancer cells and mammospheres were separated on 10% SDS-PAGE and transferred to a polyvinylidene fluoride (PVDF) membrane (Millipore, Burlington, MA, USA). Blots were blocked in 5% skimmed milk in 1XPBS-Tween 20 at room temperature for 60 min and then overnight at 4 °C with primary antibodies. The primary antibodies were PAK1, JAK2, Stat3, p65, lamin B, phospho-Stat3 (Cell Signaling), and β-actin (Santa Cruz Biotechnology). After washing, blots were detected with IRDye 680RD and 800W secondary antibodies, and signals were determined by using an ODYSSEY CLx (LI-COR, Lincoln, NE, USA). 

### 4.13. Electrophoretic Mobility Shift Assays (EMSA)

Nuclear proteins were prepared as described previously [40]. The EMSA for Stat3 and PAK1 binding was performed using an IRDye 800 Stat3 oligonucleotide (LI-COR) for 30 min at room temperature. Samples were run on a nondenaturing 6% PAGE, and EMSA data were captured by an ODYSSEY CLx (LI-COR). EMSA supershifts were analyzed by incubating nuclear proteins with PAK1 (Bethyl Lab, Montgomery, TX, USA) and pStat3 (Cell Signaling) antibodies for 30 min before the addition of the IRDye 800 Stat3 oligonucleotide.

### 4.14. Chromatin Immunoprecipitation (ChIP) and Re-ChIP Assay

The ChIP assay was analyzed as previously described [40]. A total of 1 × 10^6^ breast cancer cells were fixed with 1% formaldehyde for 15 min at 37 °C to crosslink histone- and DNA-binding proteins to DNA and washed three times with 1X PBS (phosphate buffered saline). Breast cancer cells were sonicated, and proteins were immunoprecipitated with PAK1 and pStat3 antibodies. The immunoprecipitated samples were washed and eluted from the beads. The purified DNA was amplified. The specific primers for ChIP and Re-ChIP were primers for the *IL-6* promoter (Appendix A). Re-ChIP was performed by using a Re-ChIP assay kit (Active Motif, Carlsbad CA, USA). For Re-ChIPs, first ChIP using an anti-Stat3 antibody (or anti-PAK1 antibody) was followed by a second ChIP assay using a PAK1 antibody (or Stat3 antibody).

### 4.15. Quantitative Measurement of Human IL-6 Using the BD^TM^ CBA Human Inflammatory Cytokines Assay Kit

The IL-6 measurement was performed as described previously [38]. Mammosphere derived from were cultured for five days and incubated with ivermectin (10 μM) or DMSO for 36 h. IL-6 concentration of cultured media were assayed by using BD^TM^ Cytometric Bead array (CBA) human inflammatory cytokines assay kit, by following the manufacturer’s protocol (BD). We added 50 μL of assay bead, 50 μL of cultured media or standard and 50 μL of PE-labeled antibodies to each set of sample tubes. The samples were incubated at room temperature in the dark for 3 h. The samples were washed with 1 mL washing buffer and centrifuged. After discarding the supernatant, the pellet was resuspended in 300 μL washing buffer and analyzed by flow cytometry (Accuri C6, BD).

### 4.16. In Vivo Mouse Experiment

The mouse experiments were performed as described previously [41]. Animal care and animal experiments were approved and conducted in accordance with the guidelines of the Jeju National University Animal Care and Use Committee (JNU-IACUC; Approval Number 2017-003). BALB/C nude mice (five weeks old) were from OrientBio (Seoul, Korea) and kept in a mouse facility for seven days. Five male BALB/C nude mice were injected with wild-type and PAK1-knockout HAP1 cells. Tumor volumes were measured using the formula: volume = (width^2^ × length)/2. 

### 4.17. Statistical Analysis

All data from three independent experiments are shown as the mean ± standard deviation (SD). Data were analyzed using one-way ANOVA. A p-value less than 0.05 was considered statistically significant.

## 5. Conclusions

We show that inhibition of PAK1 inhibits cell proliferation and CSC formation, and tumor growth is significantly reduced in a PAK1-knockout xenograft model. PAK1 physically interacts with JAK2, and JAK2 inhibitor treatment also inhibits mammosphere formation. PAK1 interacts with Stat3, and PAK1 and Stat3 colocalize in the nucleus. Using EMSA, ChIP, and reporter assays, we show that the PAK1/Stat3 complex binds to the promoter and regulates the transcription of *IL-6*. PAK1 dysregulation inhibits CSC formation by reducing pStat3 levels and extracellular IL-6 levels. Our results reveal that JAK2/PAK1 dysregulation inhibits the Stat3 signaling pathway and CSC formation. The PAK1/Stat3 complex regulates the *IL-6* gene. PAK1/Stat3 signaling regulates CSC formation, and PAK1 may be an important target for treating breast cancer.

## Figures and Tables

**Figure 1 cancers-11-01527-f001:**
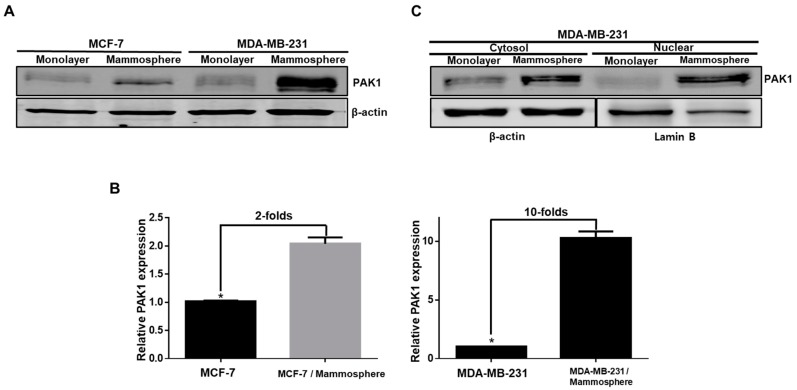
Analysis of PAK1 protein levels in breast cancer stem cells (CSCs). (**A**) Expression levels of PAK1 protein in breast cancer and CSCs. The expression levels of PAK1 were determined in breast cancer and CSCs by immunoblot using a PAK1 antibody and β-actin. (**B**) Relative expression of PAK1 in breast cancer and CSCs was estimated by densitometry. (**C**) Expression levels of cytosolic and nuclear PAK1 protein in breast cancer and CSCs derived from MDA-MB-231 cells. Cytosolic and nuclear PAK1 expression were analyzed in breast cancer and CSCs by immunoblot analysis using a PAK1 antibody. The data are presented as the mean ± SD; n = 3; * *p* < 0.05 vs. control.

**Figure 2 cancers-11-01527-f002:**
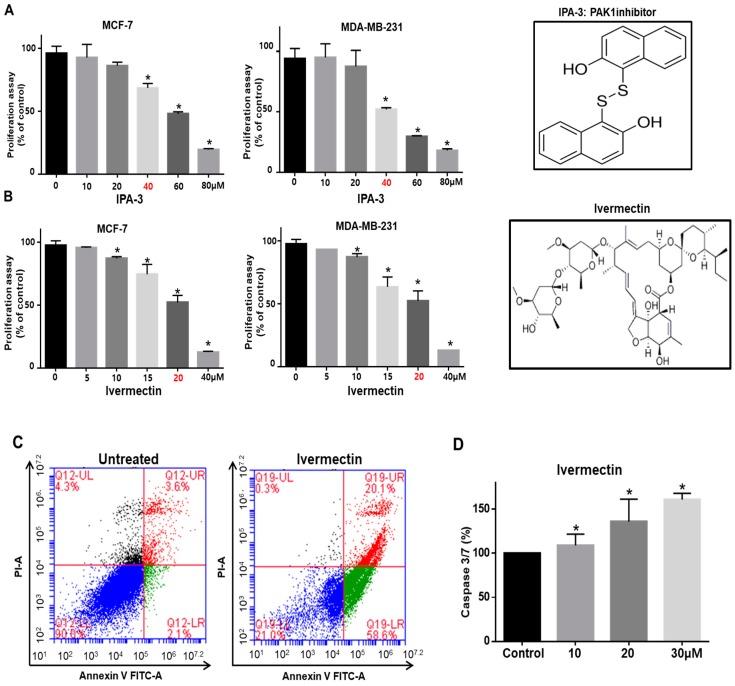
PAK1 inhibition blocks breast cancer hallmark processes. (**A**,**B**) Chemical structure of IPA-3 and ivermectin, and the effect of IPA-3 and ivermectin on the proliferation of breast cancer cells. Cancer cells were incubated with IPA-3 and ivermectin for 24 h. The antiproliferation effect of IPA-3 and ivermectin was determined by using an MTS (3-(4,5-dimethylthiazol-2-yl)-5-(3-carboxymethoxyphenyl)-2-(4-sulfophenyl)-2H-tetrazolium) assay. (**C**) Ivermectin induced apoptosis in cancer cells at the indicated concentration (20 μM). Apoptotic cells were determined using Annexin V/PI staining. (**D**) The caspase3/7 activity of cancer cells was determined using a Caspase-Glo 3/7 assay kit (Promega). The data are presented as the mean ± standard deviation (SD); n = 3; * *p* < 0.05 vs. the control. (**E**) Effect of IPA-3 and ivermectin on cancer cell migration. The migration of cancer cells with or without IPA-3 and ivermectin was photographed at 0 and 18 h. (**F**) Effect of IPA-3 and ivermectin on colony formation. The one thousand dissociated cancer cells were incubated with IPA-3 and ivermectin for 1 week. Representative images were collected. The data are presented as the mean ± SD; n = 3; * *p* < 0.05 vs. control. (**G**) PAK1 expression was analyzed in wild-type and PAK1-knockout HAP1 cells by immunoblot using a PAK1-specific antibody. (**H**,**I**) Wild-type and PAK1-knockout HAP1 cell tumor growth in immunodeficient nude mice. The human PAK1-knockout HAP1 cell inhibited tumor growth. Wild-type and PAK1-knockout HAP1 cells (5 × 10^6^ cells) were injected into nude mice subcutaneously. After seven weeks, images were captured with a camera. Tumor volumes were measured using a caliper after seven weeks. * *p* < 0.05 vs. control. Representative images were captured after five weeks. (**J**) PAK1 expression in tumors derived from wild-type and PAK1-knockout HAP1 cells was analyzed by western blot assay using a PAK1-specific antibody.

**Figure 3 cancers-11-01527-f003:**
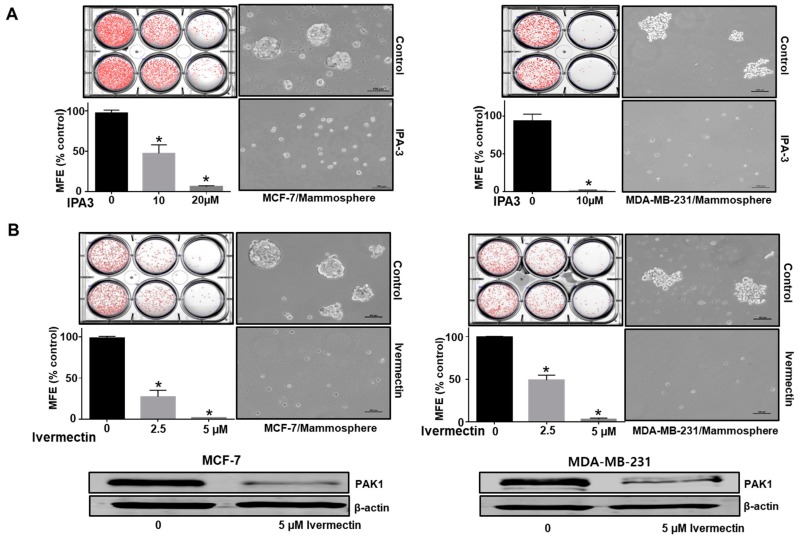
PAK1 induces mammosphere formation and regulates hallmark breast CSC processes. The PAK1 inhibitor IPA-3 and the PAK1 inactivator ivermectin were included in mammosphere induction cultures for one week. (**A**,**B**) The effect of IPA-3 and ivermectin on mammosphere formation. Primary mammospheres were treated with IPA-3 (10 and 20 µM) or ivermectin (2.5 and 5 µM). Images were captured by microscopy. (**C**,**D**) Effect of PAK1 on mammosphere formation assessed using PAK1 siRNA. PAK1-silenced cells were seeded in ultralow plates and cultured for seven days. Images were captured by microscopy at 10× magnification. The data are presented as the mean ± SD; n = 3; * *p* < 0.05 vs. control. (**E**) Effect of ivermectin, which inactivates PAK1, on the breast CSC hallmark CD44^+^/CD24^−^-positive cell population. The ivermectin (5 μM)-treated CD44^+^/CD24^−^-cell population was analyzed by flow cytometry. Ten thousand cells were collected. Gating was based on binding of the control antibody (red cross). (**F**) Effect of ivermectin on ALDH-positive cells. The breast cancer cells were analyzed using an ALDEFLUOR assay. The right panel shows ALDH-positive cancer cells with N, N-diethylaminobenzaldehyde (DEAB), and the left panel represents ALDH-positive cancer cells without DEAB. The ALDH-positive population was gated in a box (red dotted line box).

**Figure 4 cancers-11-01527-f004:**
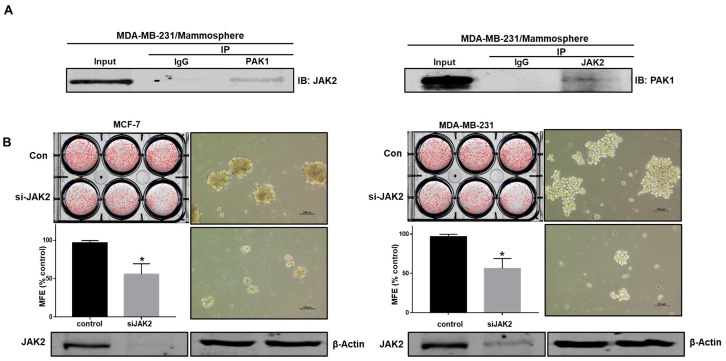
PAK1 interacts with JAK2 and induces mammosphere formation in breast cancer. (**A**) PAK1 and JAK2 were immunoprecipitated from breast cancer cell total protein with anti-PAK1 or anti-JAK2 antibodies. Western blotting was performed using the indicated antibodies. (**B**) Effect of JAK2 on the formation of mammospheres by cancer cells assessed using siRNA against JAK2. PAK1-silenced cancer cells were incubated in ultralow six-well plates for one week in CSC medium. Images were captured by microscopy at 10× magnification (scale bar; 100 μm). (**C**,**D**) Effect of TG101209 on cell proliferation and mammosphere formation. Breast cancer cells were cultured with TG101209 (from 0 to 40 µM) for 24 h. Primary mammospheres were incubated with TG101209 (2.5 and 5.0 µM) or control. The data are presented as the mean ± SD; n = 3; * *p* < 0.05 vs. control. (**E**,**F**) Effect of TG101209 on total levels of PAK1 protein. Cytosolic and nuclear PAK1 protein under TG101209 (2.5 and 5.0 µM) or dimethyl sulfoxide (DMSO) treatment conditions. (**G**) Immunofluorescence analysis of PAK1 expression in breast cancer cells under TG101209 treatment conditions.

**Figure 5 cancers-11-01527-f005:**
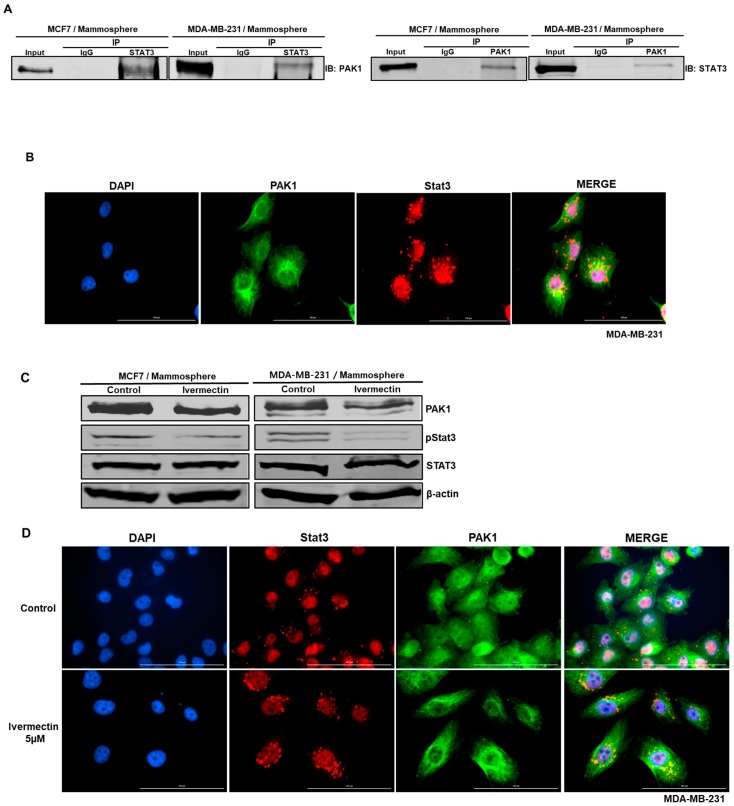
PAK1 physically interacts with Stat3, PAK1 and Stat3 colocalize in the nucleus and PAK1 inactivation induces Stat3 dephosphorylation. (**A**) Total protein from breast cancer cells was immunoprecipitated with anti-PAK1 or anti-Stat3 antibodies, followed by immunoblotting using anti-Stat3 or anti-PAK1 antibodies. (**B**) Immunofluorescence analysis of PAK1 (green) and Stat3 (red) expression in breast cancer cells using the indicated antibodies. Images were captured by microscopy at 10× magnification. (**C**) Effect of ivermectin on the pStat3 level. (**D**) Immunofluorescence analysis of PAK1 (green) and Stat3 (red) expression levels in breast cancer cells after ivermectin treatment. Images were captured by microscopy at 10× magnification (scale bar = 100 μm).

**Figure 6 cancers-11-01527-f006:**
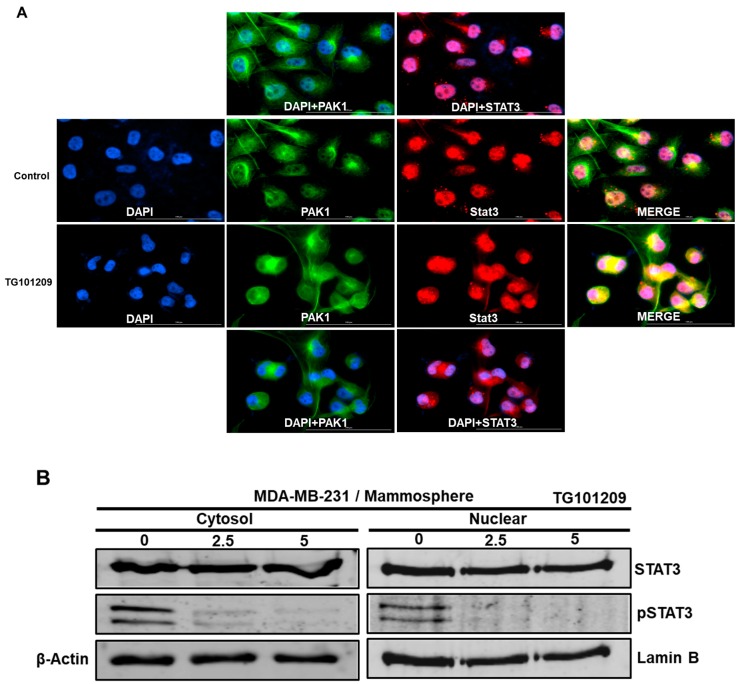
Effect of a JAK2 inhibitor on nuclear PAK1 and Stat3 levels and on the colocalization of PAK1 and Stat3. (**A**) Immunofluorescence analysis of PAK1 (green) and Stat3 (red) expression in breast cancer cells using anti-PAK1 and anti-Stat3 antibodies after TG101209 treatment. Images were captured by microscopy at 10× magnification. (**B**) Effect of TG101209 on cytosolic and nuclear pStat3 protein levels.

**Figure 7 cancers-11-01527-f007:**
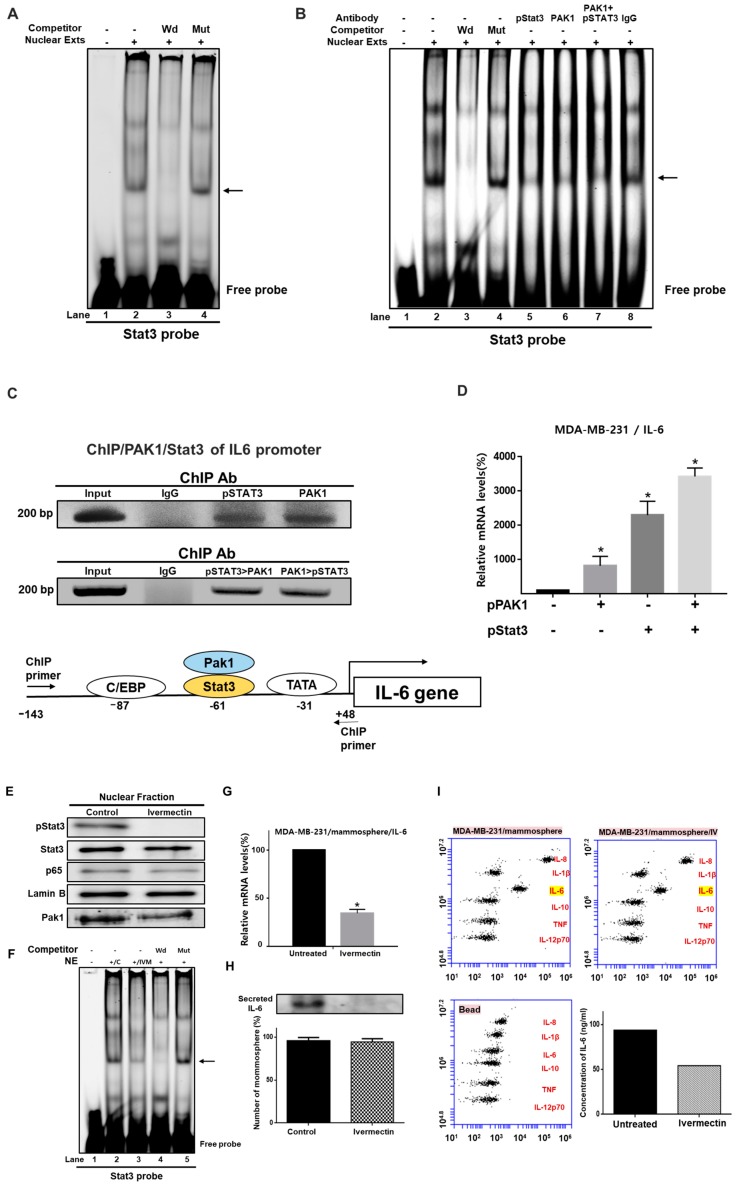
PAK1 affects *IL-6* gene expression through Stat3-mediated regulation. (**A**) Electrophoretic mobility shift assay (EMSA) using mammosphere nuclear proteins. Nuclear proteins were incubated with an IRDye 800-Stat probe at room temperature and separated by 6% native PAGE. Lane 1: stat probe; lane 2: nuclear proteins with stat probe; lane 3: 10-fold concentration of the self-competitor oligo; lane 4: nuclear proteins incubated with a 10-fold concentration of the mutated-Stat3 probe. The arrow indicates the DNA/Stat3 complex in mammosphere nuclear lysates. (**B**) EMSA analysis of PAK1 and Stat3 binding to the Stat3 DNA-binding sequence. Lane 1: Stat probe only; lane 2: nuclear proteins with Stat probe; lane 3: 10-fold concentration of the self-competitor oligo; lane 4: 10-fold concentration of the mutated Stat probe; lane 5: pStat3 antibody; lane 6: PAK1 antibody; lane 7: pStat3 and PAK1 antibody; lane 8: IgG. The arrow indicates the Stat3/PAK1/DNA complex in nuclear proteins. (**C**) ChIP (chromatin immunoprecipitation) analysis of Stat3 and PAK1 recruitment to the *IL-6* gene promoter region in breast cancer cells. Double-ChIP analysis of the Stat3-PAK1/DNA complex binding to the human *IL-6* gene promoter in breast cancer cells. The first ChIP was performed with an anti-pStat3 or anti-PAK1 antibody, followed by a second ChIP with an anti-PAK1 or anti-pStat3 antibody, respectively. Input represents 1% of the sonicated DNA. (**D**) RT-qPCR analysis of human IL-6 mRNA levels in breast cancer cells transiently overexpressing PAK1 and Stat3 genes. The relative mRNA expression levels are presented. The data are presented as the mean ± SD; n = 3; * *p* < 0.05 vs. control vector. (**E**) Nuclear levels of Stat3 and NF-κB (p65) protein in mammospheres were examined with antibodies against Stat3 and p65. Ivermectin treatment reduced nuclear pStat3 levels in the mammospheres. (**F**) EMSA of lysates of mammospheres treated with ivermectin. Nuclear proteins were treated with an IRDye 800-Stat sequence probe and separated by 6% native PAGE. Lane 1: stat probe; lane 2: nuclear proteins with stat probe; lane 3: ivermectin-treated proteins with stat probe; lane 4: 10-fold concentration of self-competitor oligo; lane 5: 10-fold concentration of mutated Stat3 oligo. Ivermectin treatment decreases the DNA/Stat3 interaction in mammosphere nuclei. (**G**) Transcriptional levels of the human *IL-6* gene were assayed in ivermectin-treated cells using *IL-6*-specific primers. β-actin expression was used to normalize the values. The data are presented as the mean ± SD; n = 3; * *p* < 0.05 vs. control. (**H**) Immunoblot analysis of culture medium from mammospheres using an IL-6 antibody and equal numbers of mammospheres as a control. Ivermectin treatment reduced extracellular IL-6 levels in mammosphere culture medium. (**I**) The cytokine profiles of conditioned media from mammosphere cultures were determined with cytokine-specific antibodies and cytokine beads. Ivermectin reduced extracellular IL-6 levels in the mammosphere cultures.

**Figure 8 cancers-11-01527-f008:**
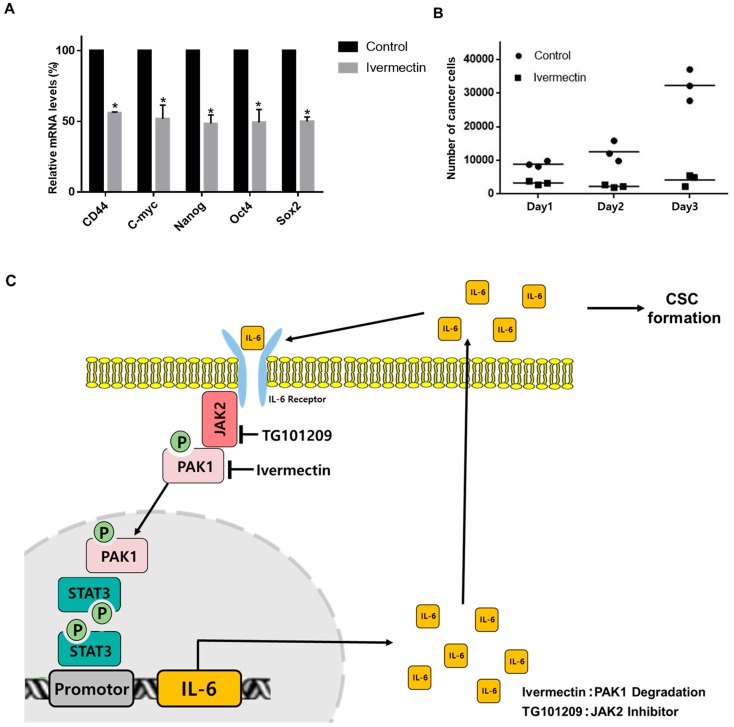
The effects of ivermectin, a PAK1 inactivator, on CSC loads in breast cancer. (**A**) Transcriptional levels of Sox2, Oct4, CD44, c-Myc, and Nanog genes were analyzed in ivermectin-treated mammospheres using specific primers. β-actin was used as a control. (**B**) Effect of ivermectin on mammosphere growth. Ivermectin treatment prevents mammosphere proliferation. Ivermectin-treated mammospheres were dissociated into single cells and plated in 6-well plates in equal numbers. Twenty-four hours after plating, the cells were counted. Two and 3 days later, the cells were counted. The data are presented as the mean ± SD; n = 3; * *p* < 0.05 vs. control. (**C**) The proposed model for CSC formation by JAK2/PAK1/Stat3/IL-6. IL-6-induced JAK2 and PAK1 interaction. JAK2 inhibitor (TG101209) and PAK1 inactivator (ivermectin) treatment blocks the Stat3 signaling pathway and *IL-6* gene expression and inhibits CSC formation. TG101209 and ivermectin treatment reduces extracellular IL-6 and Stat3 signaling. Extracellular IL-6 induces the conversion of cancer cells to CSCs. PAK1/Stat3 regulates CSC formation through Stat3 signaling and extracellular IL-6.

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
