# Peer review of "The PAK1-Stat3 Signaling Pathway Activates IL-6 Gene Transcription and Human Breast Cancer Stem Cell Formation"

_cancers, 2019, doi:10.3390/cancers11101527_

Round 1

Reviewer 1 Report

The manuscript by Kim et al., investigated the role of PAK1-STAT3 in the activation of IL-6 in breast cancer stem cell formation. By using cell lines study, they demonstrated that PAK1 is overexpressed in two mammosphere derived from breast cancer cell lines. Pharmcological and genetic suppression of PAK1 inhibited proliferation, migration, colony formation and mammosphere formation and stem cell population. Further analysis found that PAK1 activates STAT3 and subsequent activation of IL-6 by binding to the promoter of IL-6. Although this manuscript is potentially interesting, it is basically a cell line study lacking experiments from clinical samples. Furthermore, some of the experimental results are not very convincing.

Specific comments:

Fig 2F, tumor xenograft model, the growth curve of the tumor has to be shown; Figure 3E, the scatter plot from flow cytometry analysis of CD44 and CD24 is not convincing. The current result is due the interference between FITC and PE thus compensation need to be performed. The authors may also consider using different fluorescent dyes which has less overlapping. Anyhow, this experiment has to be repeated. Please refer to Al-Hajj et al., PNAS 2003 for the typical scatter plot of flow cytometry analysis of CD44 and CD24 in breast cancer. Page 11 and Figure 7B, the authors stated that major supershifted band was observed using anti-STAT3 (7B, lane 5) and anti-PAK1 antibody (7B, lane 6). However, the EMSA results did show an obvious band shift. It has to be repeated. To further demonstrate if PAK1 or STAT3 is being recruited to IL-6 promoter, the authors should perform PAK1 or STAT3 knockdown and perform ChIP-PCR to examine the binding of these two protein to IL-6 promoter. To be more clinical relevance, the authors should examine PAK1 expression mammosphere isolated from breast cancer patient samples. The authors should also examine the correlation of PAK1 and IL-6 in breast cancer patient samples from the publicly available dataset such as TCGA. Expression of IL-6 in conditioned medium has to be repeated by using ELISA assay. Please double check the histogram of Figure 7H. Discussion has to be elaborate and expand to include more references on the role of PAK1 in cancer stemness.

Reviewer 2 Report

In this manuscript by Kim, et al, the authors describe a signaling pathway involving IL-6, JAK2, PAK1 and STAT3 that promotes a stem cell phenotype, namely mammosphere formation, in breast cancer cell lines. The role of IL6/JAK2/STAT3 is well established, so what is new here is the involvement of PAK1. The authors use a variety of complementary methods to demonstrate the interactions of the various elements of the pathway. The data are generally sound and the paper is easy to read and understand. A few specific concerns and suggestions are outlined below.

Figure 1: It is somewhat confusing to refer to cell lines growing in standard monolayer as “cancer.” Would suggest using “monolayer” and “mammosphere” for greater clarity.

Figure 2A,B: The doses required are very high, much higher that what is required to see effects on mammospheres. Please comment.  Also, the authors need to measure cell death, such as with a specific apoptosis assay. With these doses it is possible that the cells are not proliferating because they are dead.

Figure 2E-H: Why use HAP1 cells? These are not breast cancer cells. I’m sure there are several breast cancer cell lines that are amenable to knockout with siRNA or CRISPR/Cas9, and that grow in vivo. Also, the box plot in G needs a better description and a title for the y axis.

Figure 3E: The flow plots look like there may be a compensation issue between FITC and PE.

Figure 3F: How was the positive gate established? The percents are so small that a tiny adjustment of the gate could have a big effect on the results. Also how many times were these studies repeated?

Figure 4A: For the blot on the right, the third lane looks overexposed compared to the others. Were all three lanes on the same gel and exposed for the same time?

Figure 4C,D: The y axes need titles.

Figure 5D: The immunofluorescence images do not look obviously different to me. Also, in the methods, please describe the controls used to establish the specificity of the antibodies. For example, did you stain slides with the secondary antibody only? These would be useful to show in the supplement.

Figure 7D: Please comment on whether the breast cancer cells need to be stimulated with something to induce phosphorylation of PAK1 and STAT3.

Line 296 pertaining to Figure 8A: It is not quite accurate to call the used for 8A “CSCs.” They are mammospheres, which may contain CSCs.

Figure 8C: Your model proposes that IL-6 stimulation is required to induce the signaling pathway, but this was not specifically demonstrated. It would be easy to test with an anti-IL6 receptor antibody, for example.

Author Response

In this manuscript by Kim, et al, the authors describe a signaling pathway involving IL-6, JAK2, PAK1 and STAT3 that promotes a stem cell phenotype, namely mammosphere formation, in breast cancer cell lines. The role of IL6/JAK2/STAT3 is well established, so what is new here is the involvement of PAK1. The authors use a variety of complementary methods to demonstrate the interactions of the various elements of the pathway. The data are generally sound and the paper is easy to read and understand. A few specific concerns and suggestions are outlined below.

Figure 1: It is somewhat confusing to refer to cell lines growing in standard monolayer as “cancer.” Would suggest using “monolayer” and “mammosphere” for greater clarity.

→ The reviewer’s point is well taken. We changed cancer to monolayer of Figure 1.

Figure 2A,B: The doses required are very high, much higher that what is required to see effects on mammospheres. Please comment. 

→ Before, our group had screened anti-CSCs compounds using the NICE program. To find more specific CSCs inhibitor, we screened inhibitor that kills a breast cancer at higher concentration and kill breast CSCs at lower concentration. Our group published a paper using (Choi HS, et al. Screening of breast cancer stem cell inhibitors using a protein kinase inhibitor library. Cancer Cell Int. 2017 Feb 13; 17:25).

Also, the authors need to measure cell death, such as with a specific apoptosis assay. With these doses it is possible that the cells are not proliferating because they are dead.

→ The reviewer’s point is well taken. We added new Figure 2C, D containing apoptosis and caspase3/7 assay.

Figure 2E-H: Why use HAP1 cells? These are not breast cancer cells. I’m sure there are several breast cancer cell lines that are amenable to knockout with siRNA or CRISPR/Cas9, and that grow in vivo.

→ We used human PAK1-knockout HAP1 cell edited by CRISPR/Cas for our experiment as human PAK1-knockout HAP1 can induce tumor. Our group want to get PAK1-knockout breast cancer cell line using siRNA or CRISPR/Cas9. But we can’t make PAK1-knockout breast cancer cell due to several reasons. We selected human PAK1-knockout HAP1 cell of horizon company.

Also, the box plot in G needs a better description and a title for the y axis.

→We added a new sentence, “human PAK1-knockout HAP1 cell inhibited tumor growth” at Figure 2G and a title for the y axis at Figure 2G (new one, Figure 2I).

Figure 3E: The flow plots look like there may be a compensation issue between FITC and PE.

→ The Reviewer’s comment is well taken. We added new Figure 3E.

Figure 3F: How was the positive gate established? The percents are so small that a tiny adjustment of the gate could have a big effect on the results. Also how many times were these studies repeated?

→We established positive gate using Specific ALDH inhibitor, DEAB and adjusted gating box at near 0% as reference. ALDH inhibitor treatment without ivermactin reduced ALDH positive cell population from 1.6 % to 0.4 %. Our assay system is working. Ivermectin treatment reduced ALDH positive cell population from 1.6 % to 0.8%. We added a box (red dot line box) for clear data presentation at Fig.3 legend. Ivermectin reduced CSCs population. We repeated experiment three time

Figure 4A: For the blot on the right, the third lane looks overexposed compared to the others. Were all three lanes on the same gel and exposed for the same time?

→ The Reviewer’s comment is reasonable. We did not overexpose blot and western blot on one gel containing three samples. We thought that antibody problem or co-IP condition occurred the event.

Figure 4C, D: The y axes need titles.

→ The Reviewer’s comment is well taken. We added titles at y axis of Figure 4C, D.

Figure 5D: The immunofluorescence images do not look obviously different to me. Also, in the methods, please describe the controls used to establish the specificity of the antibodies. For example, did you stain slides with the secondary antibody only? These would be useful to show in the supplement.

→ The Reviewer’s comment is reasonable. We used same antibody at Figure 4G, Figure5 B, D and Figure 6A. We added a clear Figure 5D.

Figure 7D: Please comment on whether the breast cancer cells need to be stimulated with something to induce phosphorylation of PAK1 and STAT3.

→ Breast does not stimulate Stat3 and PAK1 and they are constitutive activation of Stat3 through the IL6/JAK/STAT3 pathway initiated by the binding of IL6 family of cytokines and constitutive p21-activated Kinase (PAK) Activation in Breast Cancer Cells in breast cancer cells.

Reference:

Constitutive activation of STAT3 in breast cancer cells: A review. Int J Cancer. 2016 Jun 1; 138(11): 2570–2578

Constitutive p21-activated Kinase (PAK) Activation in Breast Cancer Cells as a Result of Mislocalization of PAK to Focal Adhesions. Mol Biol Cell 2004 Jun; 15(6): 2965–2977.

Line 296 pertaining to Figure 8A: It is not quite accurate to call the used for 8A “CSCs.” They are mammospheres, which may contain CSCs.

→ The Reviewer’s comment is correct. We changed the words, breast CSCs (Figure 8A) into mannospheres (Figure 8A).

Figure 8C: Your model proposes that IL-6 stimulation is required to induce the signaling pathway, but this was not specifically demonstrated. It would be easy to test with an anti-IL6 receptor antibody, for example.

→ The Reviewer’s comment is reasonable. Breast does not stimulate Stat3 and they are constitutive activation of STAT3 through the IL6/JAK/STAT3 pathway initiated by the binding of IL6 family of cytokines.

Reference:

(1)Constitutive activation of STAT3 in breast cancer cells: A review. Int J Cancer. 2016 Jun 1; 138(11): 2570–2578. Signaling through the IL-6/JAK/STAT3 pathway initiated by the binding of IL-6 family of cytokines (i.e., IL-6 and IL-11) to their receptors have been implicated in breast cancer development (2) Regulation of Cancer Stem Cells by Cytokine Networks: Attacking Cancer's Inflammatory Roots. Clin Cancer Res.2011 Oct 1; 17(19):6125-9. Regulation of cancer stem cells by inflammatory cytokine networks. Binding of IL-6-IL-6R complex to gp130 and IL-8 to CXCR1/2 activates NF-κB pathway via Stat3 and Akt signaling. IL-6 and IL-8 production by NF-κB generates a positive feedback loop which maintains constitutive pathway activation.

Round 2

Reviewer 1 Report

The authors have addressed most of the comments raised by this reviewer, however the new figure 3E is still not convincing. It is not clear if the cells express high level of CD44 and CD24 or there is still compensation issue. The authors should perform separate single staining of CD24 and CD44 (i.e. no Ab vs anti-CD24 or anti-CD44), and the results should be presented in histogram (not scatter plot) to investigate the expression of CD44 and CD24. If CD44 and CD24 turns out to have low expression, the current results may reflect a compensation issue.

Author Response

We added a new file containing reviewer comment.

Reviewer 2 Report

Thank you for your revised manuscript. There are two points that were not addressed that are important to ensure the validity of your results.

Figure 3E still has an obvious compensation problem. Depending on the cytometer used, it is common to see signal from the FITC channel in the PE channel and vice versa. This is why the cluster of events forms a diagonal population from the lower left to the upper right. This is easily corrected by adjusting the compensation, using appropriate compensation controls. The previous comment requesting a description of the controls used to ensure the specificity of the primary antibodies for immunofluorescence was not addressed. It is standard to stain cells with the secondary antibody only, to determine the degree and pattern of non-specific signal. Please describe the controls used for the immunofluorescence studies.

Author Response

(The authors gave the same response as above.)

Round 3

Reviewer 1 Report

Finally, the authors provide a satisfactory result of flow cytometry. However, it seems to this reviewer that the cells expressed a very low level of CD24, the authors should state in the discussion that, one of the limitations of this in vitro experiment is that, the cell model being used may not reflect the actual status of breast cancer cells which express both CD24 and CD44 (Ai-Hajj et al., PNAS 2003, PMID:12629218). The authors need to state whether this mammosphere is derived from which cell lines. 

Author Response

Finally, the authors provide a satisfactory result of flow cytometry. However, it seems to this reviewer that the cells expressed a very low level of CD24, the authors should state in the discussion that, one of the limitations of this in vitro experiment is that, the cell model being used may not reflect the actual status of breast cancer cells which express both CD24 and CD44 (Ai-Hajj et al., PNAS 2003, PMID:12629218). The authors need to state whether this mammosphere is derived from which cell lines.

→The reviewer’s point is well taken. We added a new paragraphs at Discussion part as followed,

Human breast cancer cells express both CD24 and CD44 (Ref, 31), but MDA-MB-231 cancer cells expressed a high level of CD44 and a very low level of CD24. Our in vitro system, MDA-MB-231 cells and mammospheres derived from MDA-MB-231 cells, have a limitation that the cell model being used may not reflect the actual status of breast cancer cells.